# Mechanical vibration patterns elicit behavioral transitions and habituation in crawling *Drosophila* larvae

**Alexander Berne[1], Tom Zhang[1], Joseph Shomar[1], Anggie J Ferrer[1], Aaron Valdes[1], Tomoko Ohyama[2], Mason Klein[1]***

[1]Department of Physics, Department of Biology, University of Miami, Coral Gables, United States; [2]Department of Biology, McGill University, Montreal, Canada

**Abstract** How animals respond to repeatedly applied stimuli, and how animals respond to mechanical stimuli in particular, are important questions in behavioral neuroscience. We study adaptation to repeated mechanical agitation using the *Drosophila* larva. Vertical vibration stimuli elicit a discrete set of responses in crawling larvae: continuation, pause, turn, and reversal. Through high-throughput larva tracking, we characterize how the likelihood of each response depends on vibration intensity and on the timing of repeated vibration pulses. By examining transitions between behavioral states at the population and individual levels, we investigate how the animals habituate to the stimulus patterns. We identify time constants associated with desensitization to prolonged vibration, with re-sensitization during removal of a stimulus, and additional layers of habituation that operate in the overall response. Known memory-deficient mutants exhibit distinct behavior profiles and habituation time constants. An analogous simple electrical circuit suggests possible neural and molecular processes behind adaptive behavior.

**\*For correspondence:**
klein@miami.edu

## Editor's evaluation

This is a strong article due to its sophisticated behavioral analysis and modeling of behavioral output, and the system and results provide a framework for future genetic analysis examining the biological basis of sensory behaviors.

## Introduction

Animals operate in environments where complex external information is sensed, processed, and ultimately influences the likelihood of each possible behavior in their repertoire. They must distinguish relevant and irrelevant information to optimize their behavior to varied (and changing) environmental conditions (*Zucker, 1972*; *Geyer and Braff, 1987*; *Jäger and Henn, 1981*; *Rose and Rankin, 2001*; *Sasaki et al., 2001*). As a result, many animals adapt to external inputs, and sometimes retain specific stimulus information (*Duerr and Quinn, 1982*; *Rose and Rankin, 2001*). How information is translated into meaningful behavioral output is an important question in neuroscience research.

An animal that can dynamically respond to stimuli increases its chances of survival. A freely crawling insect larva in search of food, for example, can react to danger, an obstacle, or other aversive stimuli by moving or changing direction. This has been observed in behavioral analysis of chemotaxis, phototaxis, thermotaxis, and mechanosensitive avoidance (*Xiang et al., 2010*; *Zhang et al., 2013*; *Rosenzweig et al., 2008*; *Gershow et al., 2012*; *Kane et al., 2013*; *Klein et al., 2015*; *van Giesen et al., 2016*; *Ohyama et al., 2013*). Consistent exposure to a stimulus can evoke habituation, where avoidance is diminished in favor of more exploratory behaviors. More complex animals show similar

characteristics: the habituation of fly larvae exposed to non-threatening aversive odors (**Eddison et al., 2012**) or *Caenorhabditis elegans* and *Aplysia* exposed to mechanical stimuli (**Rose and Rankin, 2001**; **Stopfer and Carew, 1996**; **Rosen et al., 1979**) is seen also in mice (**Crawley, 1985**; **Belzung and Griebel, 2001**). In these examples, switching between avoidance and exploratory behaviors relies on the animal's stimulus history, so they must retain some information about that history.

The *Drosophila* larva serves as a good organism for investigating short-term retention and loss of information and how these phenomena affect behavior. The animal has a limited array of simple, discrete behaviors (crawling, turning, stopping, reversing, hunching, rolling, burrowing, etc.); it moves slowly, enabling precise observation of its body movements; many relevant neurons have been identified and characterized, and the animal is optically transparent, enabling in vivo neurophysiology; and the fruit fly has many genetic tools readily available. Studies have also noted that *Drosophila* larvae can retain olfactory stimulus information for extended periods of time (**Gerber and Stocker, 2007**; **Dubnau et al., 2001**; **Brea et al., 2014**; **Quinn et al., 1974**). Tests identifying associative olfactory learning and memory have shown that larvae maintain conditioning up to 24 hr after training, with a sharp initial decay followed by a more gradual decay in memory over time (**Tully and Quinn, 1985**). Although short-term (10–20 min) olfactory habituation has been observed (**Larkin et al., 2010**), fewer studies have sought to quantitatively characterize the habituation of *Drosophila* larvae to other types of stimuli, and precise and rapid odor delivery can be complicated (**Su et al., 2011**).

Mechanical agitation serves as a good aversive stimulus to study short-term behavior. Because the intensity and timing of vibration can be controlled (**Ohyama et al., 2013**) and can evoke context-dependent responses (**Zhang et al., 2013**; **Kim et al., 2012**), we choose here to use vibration to investigate short-term behaviors associated with information retention. Both high-force touching (**Zhang et al., 2016**) and lower-force controlled vibration (**Ohyama et al., 2015**) can be precisely controlled and delivered, and can be rapidly initiated and terminated (**Ohyama et al., 2013**). *Drosophila* exhibit avoidance responses to both types of mechanical stimuli (**Zhang et al., 2013**; **Fowler and Montell, 2013**; **Kim et al., 2012**). Rolling is a stereotyped response to noxious stimuli like high-force touching (**Hoyer et al., 2018**; **Almeida-Carvalho et al., 2017**; **Zhong et al., 2010**). Weaker forms of mechanical agitation (vibration, low-force touching) lead to milder responses like reversing and turning (**Zhang et al., 2013**; **Kim et al., 2012**; **Hwang et al., 2007**), which are the primary focus of this article.

Vibration response is also important for *Drosophila* and other insects in ecological settings. Both sound waves and substrate vibrations can indicate the presence of predators, in particular parasitoid wasps (**Zhang et al., 2013**) that inject eggs into fly larvae, which hatch and feed on host tissue (**Schlenke et al., 2007**; **Fleury et al., 2004**; **Small et al., 2012**). *Drosophila* larvae are able to respond to a wide range of frequencies (**Yagi, 1937**). Vibration also serves as the primary means of communication for many species (**Cocroft and Rodríguez, 2005**). Vibration elicits a discrete set of observable behaviors and associated neural interactions in crawling larvae. Avoidance behaviors in response to non-nociceptive vibrations, during and after stimulus delivery, are typically constructed of distinct sequences: a halting of forward motion (stop), then either a continuation of the crawl (pause), a change in forward direction (turn), or backward motion (reversal) (**Zhang et al., 2013**; **Kim et al., 2012**; **Xiang et al., 2010**; **Pulver et al., 2011**). We focus exclusively on these four behaviors in this article because the geometry of our experiments (flat surface crawling) and milder stimuli do not induce other actions that affect locomotory trajectories, and we sought to avoid stronger responses above pain thresholds.

Some aspects of the neural circuitry underlying locomotion (**Kohsaka et al., 2017**) and vibration response (**Matsuo et al., 2014**) have been characterized. The four behavioral sequences noted above are initialized by the activation of dendritic arborization neurons and chordotonal neuronal complexes lining the upper and lower portions of each larva segment (**Grueber et al., 2007**; **Cheng et al., 2010**; **Ohyama et al., 2013**). The mechanosensory transformation ends by relaying information from second-order neurons in the ventral nerve cord (VNC) to motor neurons, causing muscle contractions (**Karkali and Martin-Blanco, 2017**; **Grueber et al., 2002**; **Grueber et al., 2007**; **Ohyama et al., 2013**; **Fushiki et al., 2016**). Full circuit- and molecular-level descriptions of mechanical response remain elusive (**Tuthill and Wilson, 2016**).

The stereotyped stop and reversal behaviors in larvae differ in spontaneity, excitability, and function. Stopping behavior occurs spontaneously in the absence of a stimulus, and with increased (decreased) frequency in the presence of aversive (attractive) stimuli (**Xiang et al., 2010**; **Titlow et al., 2014**; **Pulver et al., 2011**; **Riedl and Louis, 2012**). The probability of stopping after stimulus delivery

depends on the larva's stage of neuronal development, the stimulus intensity, and the stimulus history. There is also a strong component of apparent randomness. Unlike pauses or turns, reversals rarely occur spontaneously and generally require an intense aversive stimulus (*Gjorgjieva et al., 2013*; *Eddison et al., 2012*; *Berni et al., 2012*), and thus are typically considered to be stronger avoidance than a pause or turn. Although optogenetic experiments have mapped components of the neural circuit for backward locomotion (*Clark et al., 2018*), the exact mechanism responsible for the reverse crawl motion remains unclear in *Drosophila* larvae (*Tuthill and Wilson, 2016*). On the molecular side, the regulatory protein calmodulin (CaM) functions in a larva's regulation of reversals, and spontaneous reversals occur more frequently in CaM null mutants (*Karkali and Martin-Blanco, 2017*; *Heiman et al., 1996*).

In this article, we quantitatively describe the behavioral response of *Drosophila* larvae to repeated mechanical stimulation, characterizing the onset of habituation and how habituation fades over time. First, we measure the probabilities that larvae perform each type of avoidance behavior in response to a range of vibration intensities, a characterization of sensitivity to a multidimensional stimulus. We investigate how *individual* larvae transition from performing one behavior to another between stimulus pulses, and find an almost completely one-way trend away from the strongest avoidance behaviors. Second, we characterize the onset of habituation in response to vibration pulses and extract time constants to describe both de-sensitization and a more complex re-sensitization process. Third, we characterize the response and habituation processes in known memory-deficient mutants. Finally, we use an electric circuit analogy to suggest how our behavioral results have implications for neural mechanisms behind short-term stimulus information retention and processing.

Results based on the high-resolution behavioral analysis we deploy here should set the stage for deeper understanding at the cellular and molecular levels, especially in *Drosophila*, which has a wealth of genetic tools well suited for investigating neural circuitry and neural dynamics, with a convenient modular system for introducing proteins to targeted cells (*Duffy, 2002*), imaging of activity in neurons by measuring calcium concentration (*Simpson and Looger, 2018*) or voltage (*Cao et al., 2013*), determining downstream connections in circuits (*Fosque et al., 2015*), and activating or deactivating circuit elements via optogenetics (*Klapoetke et al., 2014*). The animal is also optically accessible in vivo in the larva stage.

## Results
### Vibration response maps in 2D stimulus space

We designed and constructed a device to deliver a precisely timed sequence of pulses of mechanical vibration of specific frequency and force. An electromechanical transducer (EMT) provides sinusoidal vertical vibration, and a CCD camera records the shapes and trajectories of multiple larvae crawling on an agar gel atop the EMT's customized platform (*Figure 1A*), which is made of two metal plates together, designed to support spatially uniform stimulus delivery to all animals across the platform (see 'Materials and methods'). The instrument delivers mechanical vibration to the animals, and we describe the stimulus using two timing parameters and two intensity parameters. The time $T_{ON}$ is the duration of each vibration application, and $T_{OFF}$ is the time between the end of one vibration pulse and the start of the next. The period of the cycle we denote $T = T_{ON} + T_{OFF}$. The vertical displacement of every larva during vibration is $z(t) = A \sin 2\pi f t$, where $f$ is the frequency and $A$ the amplitude (maximum displacement). Taking a cue from engineering and materials science applications of vibration testing (*Burtally et al., 2002*; *Klein et al., 2006*), we describe intensity with both $f$ and the dimensionless peak acceleration $\Gamma \equiv A\omega^2/g$, where $\omega = 2\pi f$, and $g$ is the acceleration of gravity. A schematic of a typical stimulus is shown in *Figure 1B*, where time $t = 0$ marks the onset of the first in a series of vibrations, each counted with an index $n$ (the initial pulse labeled as $n = 0$). We use $f$ and $\Gamma$ as our parameters because materials or instruments are impacted both by the rate of vibration (especially near resonance frequencies) and by the amount of force delivered ($\Gamma$ is the peak acceleration, proportional to the force, scaled in units of $g$). We expect the same holds for biological systems. Together the four parameters ($f$, $\Gamma$, $T_{ON}$, $T_{OFF}$) fully describe the stimulus for any experiment we perform in this article.

We sought to characterize how the strength of avoidance response in crawling larvae depends on the strength of the applied vibration stimulus. In the 2D free-crawling assay employed here, we

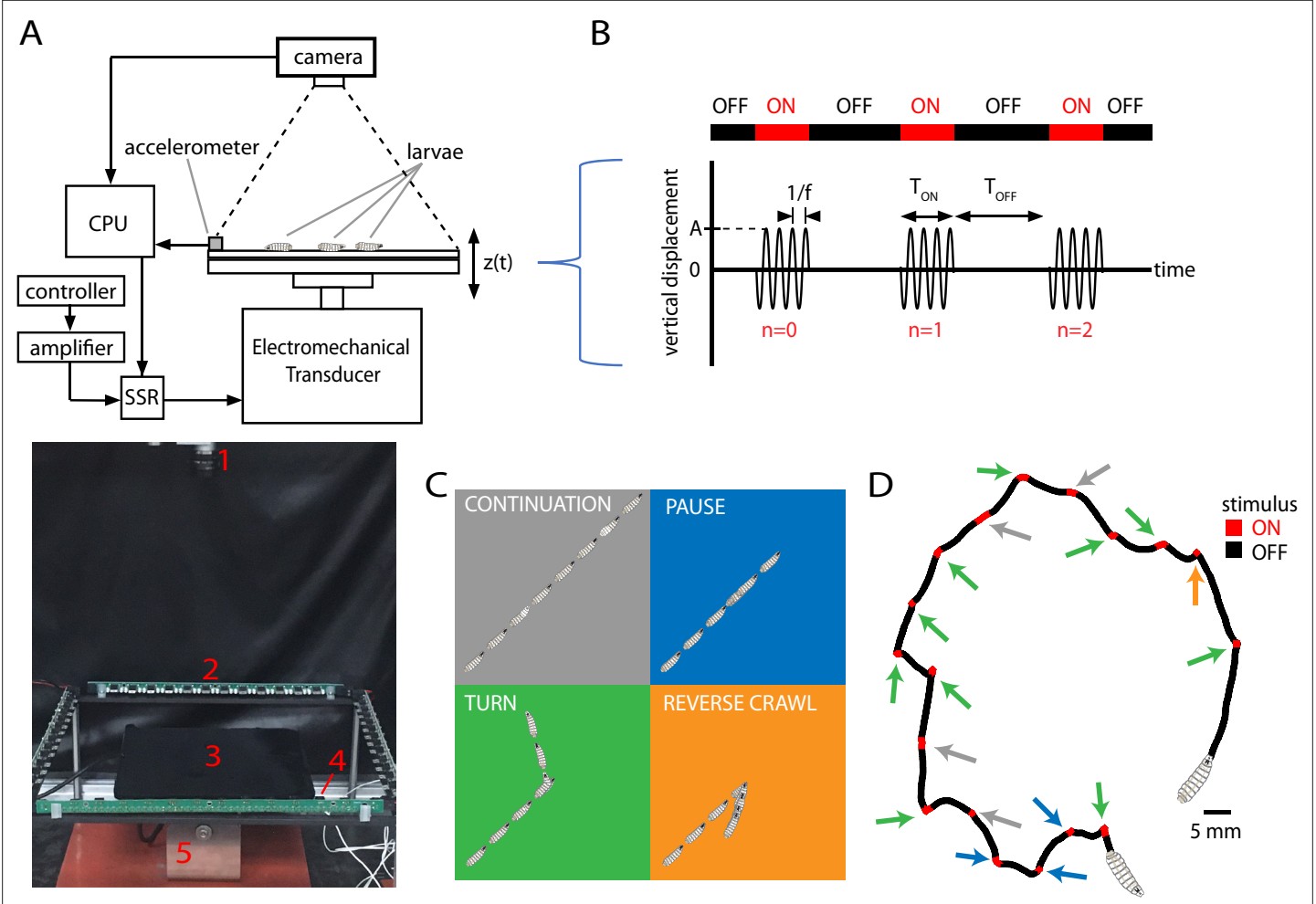

**Figure 1.** Vibration stimulus delivery and avoidance behavior classification. (**A**) Top: schematic of the experimental setup, where larvae crawl on a vertically vibrated agar gel supported by aluminum and steel plates. An electromechanical transducer provides vibration, while a CCD camera records 2D crawling of ≈ 20 red-light-illuminated animals simultaneously. See 'Methods and materials' for details. Bottom: photograph of the experimental setup, with red numbering labeling (1) camera, (2) LED array, (3) crawling substrate, (4) accelerometer, and (5) mechanical shaker. (**B**) Stimulus pattern in a typical experiment. Beginning at time $t = 0$, pulses of sinusoidal vibration are delivered for a duration of $T_{ON}$, and repeated at times $t = n(T_{OFF} + T_{ON}) = nT$, where $n$ is an integer referring to $n$th application of the stimulus. The initial vibration is referred to as the $n = 0$ stimulus, the next as $n = 1$, etc. Vibration strength is described by the frequency $f$ and the peak (dimensionless) acceleration $\Gamma$. In the top horizontal bar, red indicates stimulus ON, and black indicates stimulus OFF. (**C**) Schematic of four behavioral responses to non-nociceptive vibration: continuation (gray), pause (blue), turn (green), and reverse (orange). In each illustration, the larva crawls forward from the bottom left, and a stimulus is delivered in the center. Pictures in the sequence are equally spaced in time. (**D**) Representative trajectory of a single larva crawling for 300 s during a vibration experiment ($f = 500$ Hz, $\Gamma = 2$, $T_{ON} = 10$ s, $T_{OFF} = 20$ s). The four behaviors are indicated by arrows matching the behavior's color from (**C**). The regions with the stimulus ON appear shorter than the OFF segments because larvae come to a stop before adjusting direction, and this can take at least several seconds.

The online version of this article includes the following source code for figure 1:

**Source code 1.** Code from MATLAB that determines the behavioral state of crawling larvae.

classify larva behavioral response with four possible actions, in ascending order of avoidance strength: (1) continuing; (2) pausing; (3) turning; and (4) reverse crawling (***Figure 1C and D***). The first three behaviors occur frequently even in the absence of an aversive stimulus, whereas reversals rarely do (***Gjorgjieva et al., 2013***). Thus, we refer to continuation as 'non'-avoidance, pauses and turns as 'weak' avoidance, and reversals as 'strong' avoidance behavior. Following previous work (***Luo et al., 2010***; ***Lahiri et al., 2011***), we treat 2D larval trajectories as alternating sequences of *runs* and *reorientations*: runs are bouts of forward crawling; reorientations occur when travel speed drops near zero, asymmetric muscle contractions in segments near the head point the animal in a new direction, and

forward motion resumes. For the present classification system, we flag a 'stop' when the larva drops significantly in speed, and from there: 'pause' if forward motion resumes with a change in orientation of $\Delta\theta < 30°$, 'turn' if $\Delta\theta > 30°$, and 'reverse' if the head-pointing direction and overall velocity are in opposing directions.

In general $F_{n,ACTION}$ refers to the fraction of larvae performing 'ACTION' in response to the $n$th application of the stimulus. Response to the initial ($n = 0$) stimulus would be described by

$$
\begin{aligned}
F_{0,CONT} &= \frac{N_{CONT}}{N}, \\
F_{0,PAUSE} &= \frac{N_{PAUSE}}{N}, \\
F_{0,TURN} &= \frac{N_{TURN}}{N}, \\
F_{0,REV} &= \frac{N_{REV}}{N},
\end{aligned}
\tag{1}
$$

where $N_{CONT}$, $N_{PAUSE}$, $N_{TURN}$, and $N_{REV}$ are the number of larvae that perform a continuation, pause, turn, or reversal, respectively, and $N$ is the total number of active larvae. We also use $F_{STOP}$, the fraction of larvae that performed any kind of avoidance behavior; by definition, $F_{STOP} \equiv F_{PAUSE} + F_{TURN} + F_{REV}$. Also by definition, $F_{CONT} + F_{PAUSE} + F_{TURN} + F_{REV} = 1$.

These fractional behavioral responses are mapped to vibration conditions in $f - \Gamma$ space in *Figure 2*. In agreement with other studies indicating that reverse crawling is specifically a reaction to aversive stimuli (*Kernan et al., 1994*; *Hughes and Thomas, 2007*), our control data ($\Gamma = 0$, no vibration) shows a very small number of reversals in the $t = 0 - 2$ s time window ($F_{0,REV} = 0.03$), while larvae perform pause and turn behaviors at a baseline level with no stimulus ($F_{0,STOP} = 0.24$). During repeated vibrations for a given $\Gamma$, $f$ condition, we observed habituation: a steady decrease over time in the fraction of larvae performing the stronger avoidant reverse crawl behavior ($F_{REV}$), and in the fraction exhibiting any avoidance behavior ($F_{STOP}$), both during and between stimuli (individual plots in *Figure 2*). This suggests that larvae habituate to the presence of vibration, and that habituation does not immediately 'clear' when the stimulus turns off.

To more comprehensively understand overall habituation to vibration stimulation, we characterized how, within a population, the fraction of animals deploying each possible behavior ($F_{CONT}, F_{PAUSE}, F_{TURN}, F_{REV}$) shifts during repeated exposure to the stimulus. The fractional usage of all four behaviors over a longer time scale is shown in *Figure 2C*. In that example ($\Gamma = 2$, $f = 500$ Hz), reversal fraction $F_{REV}$ diminishes in favor of turn fraction $F_{TURN}$. To see how this fits within the larger vibration intensity parameter space, we constructed a compound graph showing fractional avoidance behavior usage during repeated vibration pulses, for 29 distinct combinations of $f$ and $\Gamma$ (*Figure 2D*). While the shift away from $F_{REV}$ appears to hold throughout $f - \Gamma$ space, many vibration settings do not cause appreciable reversal behavior at all, particularly for very low frequencies or accelerations. As a general trend, increasing vibration strength by adjusting either frequency or peak acceleration increases the fraction of both stopping and reversing larvae. We note that the relationship is not linear, but instead increasing $f$ or $\Gamma$ yields a sharper transition of behavior within the range of these two parameters explored here, where a threshold in vibration space separates reversing and non-reversing behavioral response.

To better focus our study and reduce the parameter space we need to explore, for the remaining experiments we will use vibrational strength $\Gamma = 2$ and $f = 500$ Hz, as a reliable way to study the stronger reverse crawl response, which over half the animals exhibit under those vibration conditions.

## Habituation is an essentially one-way process in individual larvae

In addition to a population-level treatment of habituation, we investigated the behavior of individuals during exposure to repeated vibration stimuli. Using recorded trajectories (positions and body contours over time) of many individual crawling larvae, we extracted behavioral sequences and noted how each animal responded to each vibration ($\Gamma = 2$, $f = 500$ Hz) in a sequence of pulses (*Figure 3A*). Each response was determined by a larva's locomotion during the first 3 s after each vibration pulse was turned on. A larva already in the middle of an action will be counted as using that action, unless forward crawling transitions into a pause, turn, or reversal during the first 3 s, in which case the new action is assigned. This window is long enough to allow for a muscle contraction cycle to finish

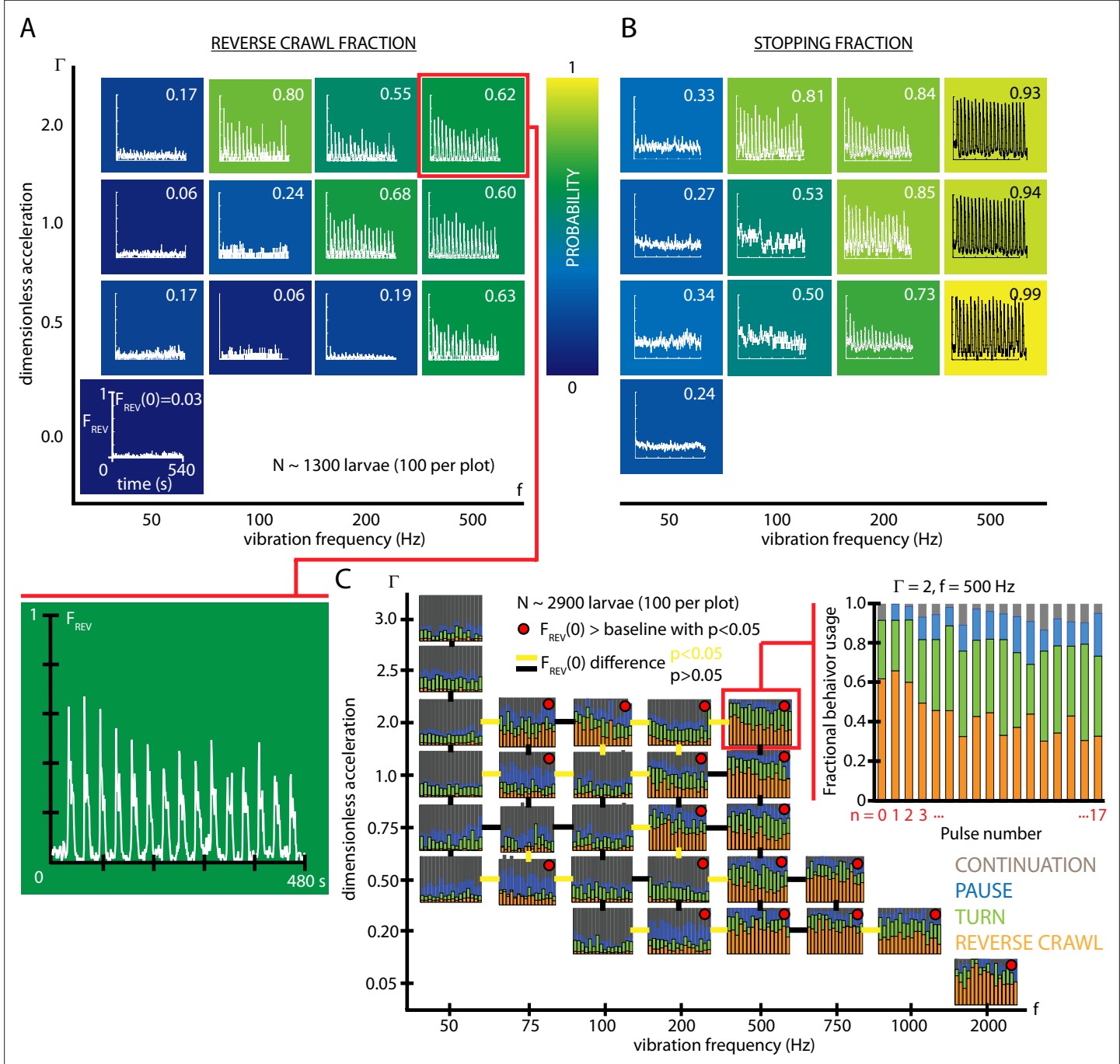

**Figure 2.** Fractional strong and weak behavioral responses depend on vibration strength. (**A**) Reversal behavior heat map. Vibration parameters were $T_{ON} = 10$ s, $T_{OFF} = 20$ s, with $\Gamma$ ranging from $0 - 2$ and frequency between 50 and 500 Hz. $F_{0,REV}$, the fraction of larvae that reverse crawl after the first ($n = 0$) vibration pulse is printed for each $f - \Gamma$ square region, alongside graphs of $F_{REV}(t)$, averaged over all experiments. Color indicates the $F_{0,REV}$ value. All graphs have the same scale in $F$ and $t$. Each ($f, \Gamma$) result is based on five experiments, each with $\approx 20$ larvae (total 1300 animals), and lasting 600 s. Note that the $f$ and $\Gamma$ axes are not on a linear scale. Uncertainties in $F_{0,REV}$ are not listed, but are $< 0.001$ for all values. (**B**) Stopping behavior heat map. From the same experiments as (**A**), but considering $F_{STOP}$, the fraction of larvae showing any avoidance behavior (pause, turn, or reversal). As vibration strength increases (along either the $f$ or $\Gamma$ axes), the fraction of avoidant larvae increases. (**C**) Inset: fractional deployment of the behavioral repertoire during habituation. $F_{REV}$ (orange), $F_{TURN}$ (green), $F_{PAUSE}$ (blue), and $F_{CONT}$ (gray) during a 3-s window after pulse initiation, as a function of the pulse number $n$. Over time the stronger avoidance behavior diminishes in favor of weaker avoidance and nonavoidance. $\Gamma = 2$, $f = 500$ Hz. Larger graph: behavioral repertoire over a range of vibration space. Fractional use of behaviors as a function of vibration pulse number ($n$) for repeated vibrations ($T_{ON} = 10$ s, $T_{OFF} = 30$ s), for many specific $f$, $\Gamma$ combinations. Each experimental condition is represented by a $F$ vs. $n$ plot, and the response of 100 larvae is averaged, for a total of 2900 animals. Lines bridging adjacent graphs indicate whether the two sets of vibration conditions

*Figure 2 continued on next page*

*Figure 2 continued*

induce a significantly different reverse crawl probability following the $n = 0$ pulse, colored either yellow for p<0.05 and black for p>0.05 (Fisher's exact test). Red dots on the graphs indicate a reverse crawl probability significantly greater than the baseline (zero vibration) response probability (Fisher's exact test).

(physical actions of the larva are on the order of 1 s), but short enough so that habituation does not become too prominent. Larger response windows do not substantively change any results in this figure or elsewhere.

Every transition (e.g., $REV \rightarrow PAUSE$) or repeat (e.g., $PAUSE \rightarrow PAUSE$) was counted and compiled to form *Figure 3B and C*, which effectively gives the probability for an individual to switch from behavior $X$ in response to one pulse to behavior $Y$ in response to either the next pulse (B) or the fifth pulse after (C).

Stronger avoidance behaviors, when not repeated, tend to switch to weaker avoidance behaviors, consistent with the population results. Of particular note is that an individual animal almost never returns to the stronger (reverse crawl) behavior after responding with a weaker one. Specifically, when comparing an assigned behavior to the behavior five pulses later, we found zero instances of transitions to reverse crawling, and zero instances of transitioning out of the continuation non-response. Thus, habituation appears to be a one-way process, at both population and individual levels, indicated

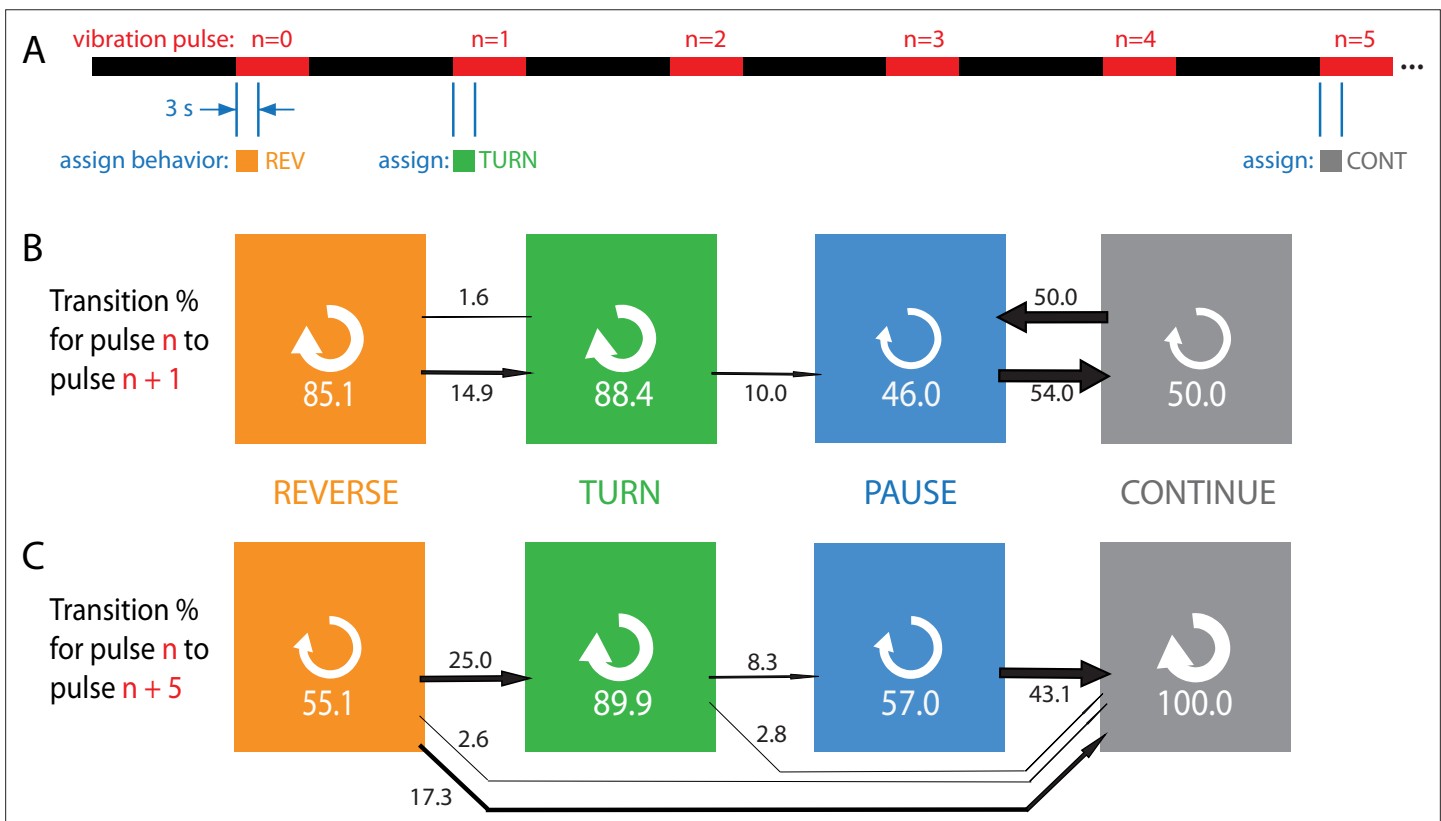

**Figure 3.** Habituation to repeated pulses is an essentially one-way process for individuals. (**A**) Schematic of the stimulus pattern and example analysis. The stimulus consisted of vibration ($f = 500$ Hz, $\Gamma = 2$) with repeated pulses of width $T_{ON} = 10$ s, repeated after $T_{OFF} = 20$ s. The behavior of each individual, in the 3 s following the onset of each vibration pulse, was assigned to one of four categories: reverse crawl (orange), turn (green), pause (blue), or continuation (gray). In the example shown, a larva reverse crawls in response to the $n = 0$ pulse, then turns in response to the next, and continues in response to the $n = 5$ pulse. (**B, C**) Behavioral transitions during repeated stimuli for individual larvae. For a given behavior observed in response to pulse $n$, the arrows represent the percentage of larvae that exhibit each of the four behaviors in response to pulse $n + 1$ (**B**) or $n + 5$ (**C**). White circular arrows represent repeating the same behavior, and the thickness of the black arrows is proportional to the fraction of animals that make the respective transition. The sum of the repeat arrows and all outgoing arrows is 100 for each behavior. Larvae were observed in five separate experiments, for a total of 107 animals making $\approx 1800$ behavioral transitions.

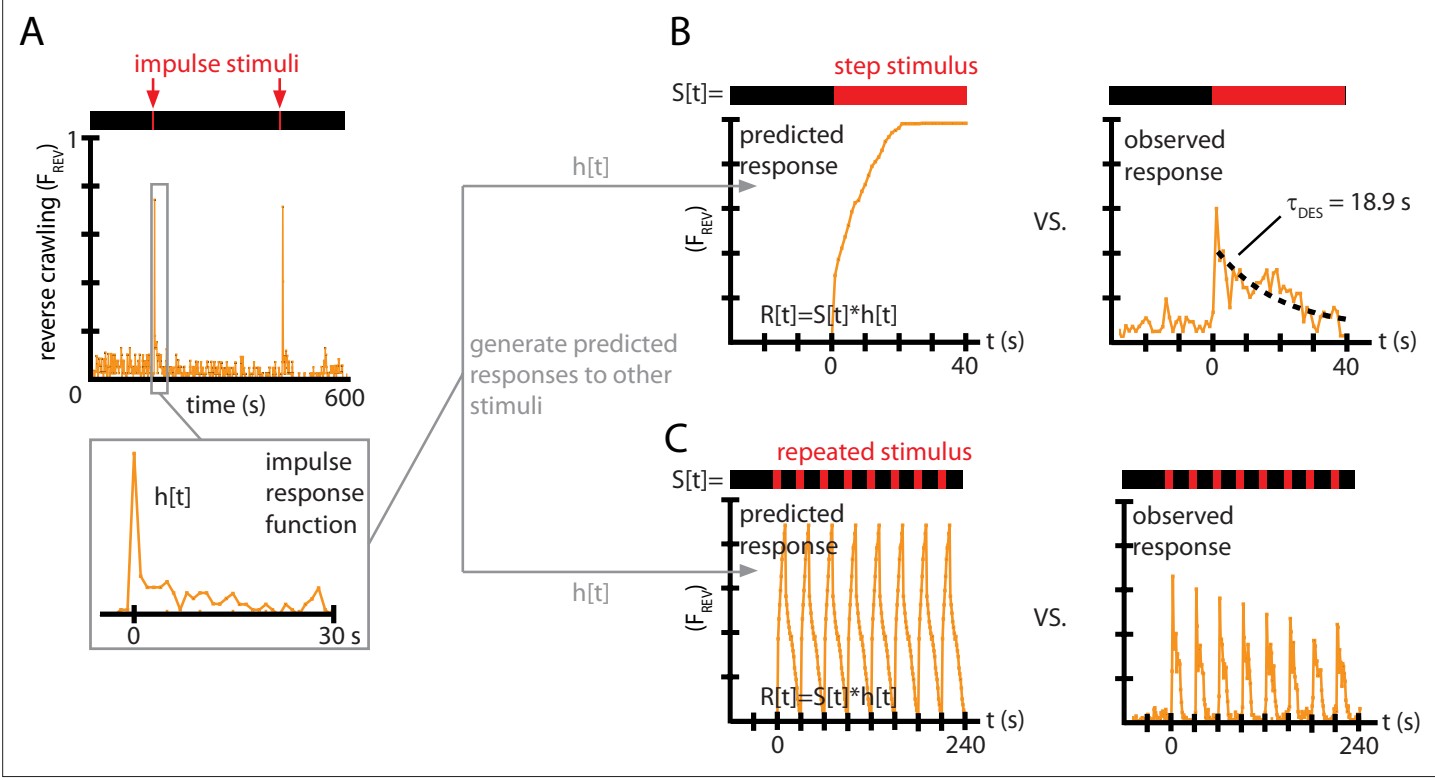

**Figure 4.** Impulse response experiments show that avoidance response to vibration is nonlinear and adaptive. (**A**) The fraction of larvae performing reverse crawling ($F_{REV}(t)$) while exposed to very short bursts of strong vertical vibration ($f = 500$ Hz, $\Gamma = 2$, $T_{ON} = 1$ s, $T = 300$ s). Inset shows a time-expanded view of the response, labeled as $h[t]$ to denote the impulse response function (IRF) used to make predictions for other stimulus inputs. (**B**) Avoidant response ($F_{REV}$) to continuous vibration, as predicted by a linear, time-invariant (LTI) model using the impulse responses $h[t]$ from (**A**) (left), and as observed empirically (right) (i.e., $T_{OFF} = 0$) with $f = 500$ Hz and $\Gamma = 2$. (**C**) Avoidant response ($F_{REV}$) to repeated pulse vibration ($f = 500$ Hz, $\Gamma = 2$, $T_{ON} = 10$ s, $T_{OFF} = 20$ s), as predicted by a linear, LTI model using the impulse responses $h[t]$ from (**A**) (left), and as observed empirically (right). The LTI calculation fails to predict the empirical behavior due to de-sensitization (**B**) and slow re-sensitization (**C**). Each plot is the average from five experiments using 20 larvae each (total 100 animals).

by the general flow of the arrows to the right in **Figure 3**, with the effect becoming more dramatic as more time elapses.

## Rapid habituation during continuous and pulsed vibration

In an attempt to more precisely understand the larva's complex behavioral response to vibrations, we turned to a signal processing method that generates a mathematical function that could predict the animal's response to any mechanical stimulus. If a system is approximately linear and time-invariant (LTI), a common technique (**Koopmans, 1995**) is to determine the system's impulse response function (IRF). In principle, this means applying a stimulus ($S$) in the form of a delta function, $S(t) = \delta(t)$, and measuring the system's response $h(t)$. That specific response function then becomes a predictive filter of behavior, such that the general response $R(t)$ to any stimulus $S(t)$ would be

$$R(t) = S(t) * h(t) = \int_{-\infty}^{+\infty} S(\tau)h(t - \tau)d\tau \tag{2}$$

We limited our scope to a single vibration intensity ($f = 500$ Hz, $\Gamma = 2$), and approximated a delta function impulse with a short sinusoidal vibration burst lasting $T_{ON} = 1$ s, with a long time between bursts ($T = 300$ s). The resulting fractional behavioral response $F_{REV}(t)$ (**Figure 4A**) shows an abrupt spike in reverse crawl behavior immediately after the vibration impulses ($t = 0$ and $t = 300$ s), followed by a slower return to baseline that takes approximately 15–20 s. We note that this impulse response form, in a sense the 'decay' of the avoidance behavior upon removal of the stimulus, is similar to the decay of olfactory conditioning memory (**Tully and Quinn, 1985**), although on a much shorter time scale.

We used this impulse response to generate predictions of the reversal behavior $F_{REV}$ under two other, distinctly different vibration pulse conditions. With the same $f$ and $\Gamma$ used to determine the IRF, we first measured response to a continuous vibration stimulus starting at $t = 0$, and then measured response to repeated pulses ($T_{ON} = 10$ s, $T_{OFF} = 20$ s). For both comparisons, we used the $F_{REV}(t)$ function from *Figure 4A* as $h(t)$. We then computed the discretized version of the convolution from *Equation 2*, $R[t] = \sum S[\tau]h[t-\tau]$, with time steps of 1 s, to generate predicted responses to the continuous vibration or to the repeated pulses, $F_{REV}(t)$.

Comparing these predictions to the empirically observed behavior (*Figure 4B and C*), we find that the LTI predictions fail in two important ways. First, in response to a continuous stimulus, larvae do not maintain their stopping or reversal rates, but instead return to baseline after $\approx 20$ s. Second, in response to the repeated pulses, not only does the avoidance behavior not continue during the entirety of the 10 s bursts, but the response at the beginning of each burst diminishes over time. This can also be observed in every representative inset graph of *Figure 2A and B* with significant initial avoidance.

Taken together, these results show that non-nociceptive vibration response in *Drosophila* larvae is not linear, and in fact shows significant signs of habituation (or de-sensitization), which we explore more comprehensively in the sections to follow.

## Re-sensitization rates increase after repeated vibration pulses

*Drosophila* larvae rapidly adapt to continuous vertical vibration, where their fractional usage of reversal and stopping behaviors returns to their baseline, no-stimulus levels (seen in *Figure 4C*). We characterize this as an exponential decay of strong avoidance behavior,

$$F_{REV}(t) = F_{0,REV} e^{-t/\tau_{des}} \quad [OFF \to ON], \tag{3}$$

where $\tau_{des}$ is the de-sensitization time constant, and $t = 0$ indicates the onset of the stimulus. Fitting an exponential to the continuous response data, we find $\tau_{des} = 18.9$ s, for wild-type larvae exposed to ($f = 500$ Hz, $\Gamma = 2$) vibration.

The fact that strong avoidance behavior (measured by $F_{REV}$) is not the same for each vibration pulse in a repeated sequence implies that larvae do not immediately reset or clear habituation to the stimulus. Thus, there is another important time constant, for re-sensitization (or de-habituation) to mechanical vibration while the stimulus is off. We describe this by

$$F_{REV}(T_{OFF}) = F_{0,REV}(1 - e^{-T_{OFF}/\tau_{res}}) \quad [ON \to OFF], \tag{4}$$

where here $t = 0$ marks the ON→OFF stimulus transition, the time $T_{OFF}$ marks the return of vibration, and $\tau_{res}$ is the re-sensitization time constant. Determining $\tau_{res}$ requires substantially more experiments than for $\tau_{des}$ because one must systematically vary $T_{OFF}$ in separate experiments to construct the shape of the function in *Equation 4*. *Figure 5A* shows the re-sensitization process for wild-type larvae exposed to ($f = 500$ Hz, $\Gamma = 2$) vibration, and we find $\tau_{res} \approx 5$ s describes de-habituation following the first vibration pulse under these conditions.

We also investigated whether the time constant $\tau_{res}$ is in fact constant over the repeated vibration pulses in a longer stimulus sequence. Using timing settings of $T_{ON} = 30$ s (sufficient for the population to habituate to its baseline $F_{REV}$ level) and a variable $T_{OFF}$, we determined separate $\tau_{res}$ at each $n = 0, 1, 2, ...$ pulse. We find (*Figure 5B and C*) that the re-sensitization rate increases dramatically: by the $n = 4$ vibration pulse, the return to the sensitivity level of the previous pulse (i.e., $F_{4,REV}/F_{3,REV} \approx 1$) happens in less than 1 s. We also note that turning off vibration does not in itself affect the fraction of larvae that perform reverse crawl behavior, although the fraction of larvae that stop does decrease temporarily (*Figure 5D*), consistent with a 'relief' period following the removal of an aversive stimulus (*Denny, 1976*).

To determine whether $\tau_{des}$, $\tau_{res}$, and ($\tau_{res}$ vs. $n$) are sufficient to explain the habituated responses to vibration stimuli, we used the three features to construct a predictive function for $F_{REV}(t)$ for a distinctly different repeated pulse stimulus input. Using ($f = 500$ Hz, $\Gamma = 2$, $T_{ON} = 10$ s, $T_{OFF} = 20$ s) as vibration conditions, we compare empirical $F_{REV}(t)$ to that predicted by the extracted constants (*Figure 5E*). The predictive function is

$$F_{REV}(t) = F_{0,REV} \cdot \sum_n \left[ 1 - e^{-T_{OFF}/\tau_{res}(n-1)} \right] \cdot e^{-(t-nT)/\tau_{des}}$$

$$\tag{5}$$

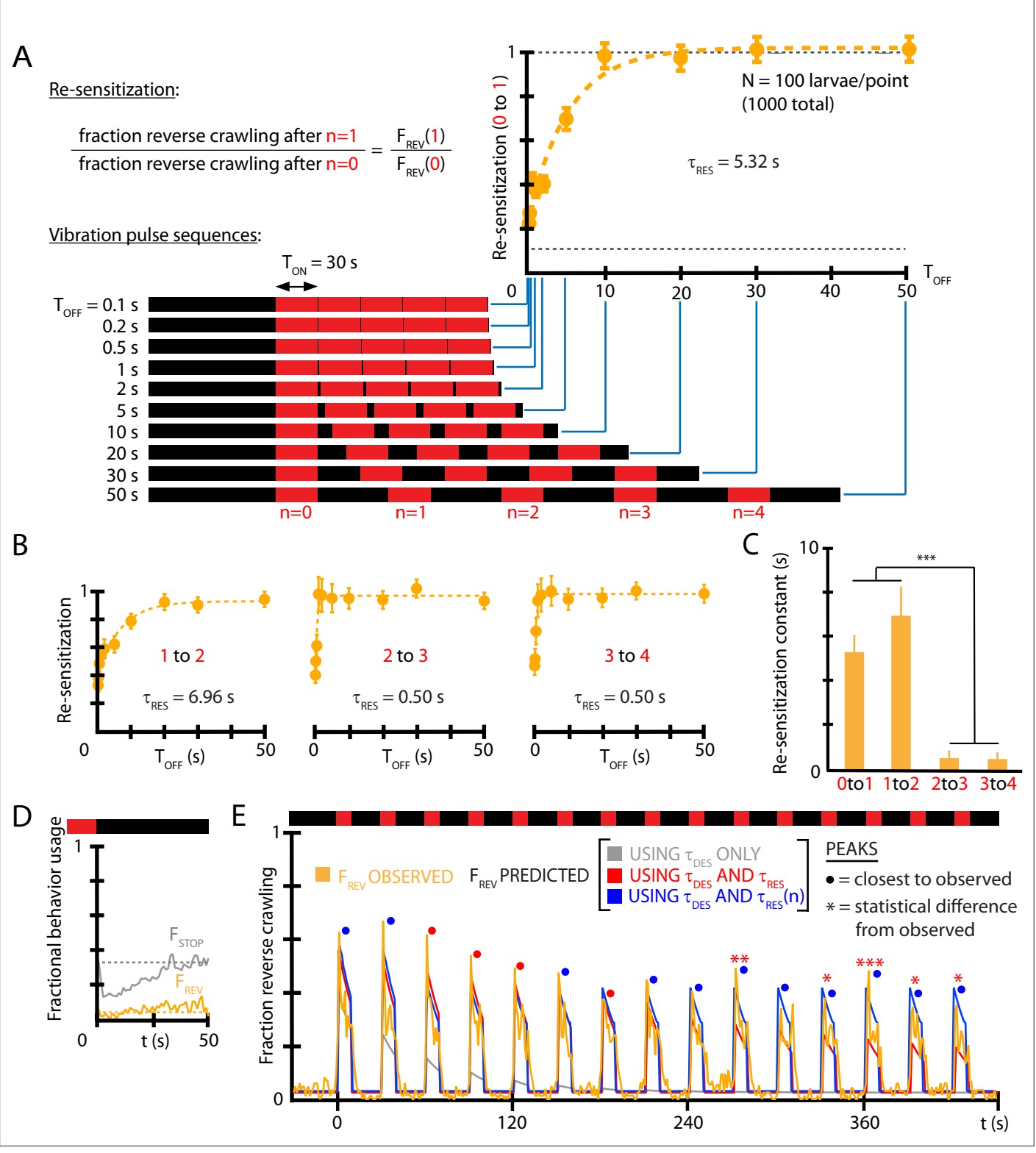

**Figure 5.** Re-sensitization after removal of mechanical stimulus depends on prior vibration pulses. (**A**) Visualization of vibration pulse sequence experiments used to determine re-sensitization to the stimulus. Time $t = 0$ indicates the start of the initial ($n = 0$) vibration pulse period, lasting $T_{ON} = 30$ s (red). The stimulus is removed between pulses for varying amounts of time $T_{OFF}$. Recovery of sensitization is determined for each pulse $n$ by computing the ratio $F_{n,REV}$ to $F_{n-1,REV}$, normalized to account for incomplete recovery for short $T_{OFF}$ times. Lower dashed line indicates baseline (no stimulus) reversal fraction. (**B**) Re-sensitization as a function of the time $T_{OFF}$, determined for the $n = 1$, $n = 2$, $n = 3$, and $n = 4$ pulses.

*Figure 5 continued on next page*

*Figure 5 continued*

Vibration intensity was $f = 500$ Hz, $\Gamma = 2$. Each data point is the average from five experiments of $\approx 20$ animals each, for a total of 1000 larvae from 50 experiments. Error bars are SEM. (**C**) Re-sensitization time constants as a function of vibration pulse number $n$. $\tau_{res}$ was determined from fits of the data in (**A**) and (**B**) (**Equation 4**). After two vibration pulses, the re-sensitization is significantly faster (***p<0.001, Student's *t*-test). (**D**) Behavioral response to the ON→OFF stimulus transition: $F_{REV}(t)$ and $F_{STOP}(t)$, where $t = 0$ indicates the stimulus OFF transition. $F_{REV}$ is unaffected. Vibration conditions ($f = 500$ Hz, $\Gamma = 2$, $T_{ON} = 50$ s, $T_{OFF} = 30$ s). Data points are the average of $F_{STOP}$ (gray) and $F_{REV}$ (orange) up to the $n = 9$ pulse. Dashed lines indicate the baseline behavior fractions while the stimulus is ON. (**E**) Comparison of habituation models with $\tau_{des}$, $\tau_{res}$, and $\tau_{res}$ vs. $n$ dependence (blue) to empirical strong avoidance behavior $F_{REV}(t)$ (orange), as well as more limited models with only $\tau_{des}$ (gray) or only $\tau_{des}$ and $\tau_{res}$ (red). Colored circle markers above each peak indicate which model was closest to the empirical peak value, while * symbols indicate whether the model peak value was significantly different than the empirical peak value (Fisher's exact test, *p<0.05, **p<0.01, ***p<0.001, gray trace not included).

The online version of this article includes the following source code for figure 5:

**Source code 1.** Code in Igor Pro for performing fits to determine desensitization and re-sensitization time constants from reversal count data.

when the stimulus is ON following the $n$th vibration pulse, and $F_{REV}(t) = 0$ when the stimulus is OFF. The predictions disagree at later times without the $\tau_{res}$ vs. $n$ dependence, but show agreement when that element is included.

Taken together, we have determined that larvae habituate and de-habituate (or de-sensitize and re-sensitize) on distinct time scales, and that re-sensitization becomes an extremely fast process after several vibration pulse repetitions, indicating an additional layer to the adaptation process. We note that although the maps of individual responses to repeated vibration pulses (**Figure 3B and C**) tend toward weaker responses over time (e.g., very rarely will a reversal follow a turn), these responses occur after habituation (vibration remaining on for 10 s) and de-habituation (while vibration is off) have both occurred, allowing individual larvae to at least partially reset. The re-sensitization mechanism characterized in **Figure 5** helps explain why the strong vibration responses repeat so often (85% for reversals).

## Memory-deficient mutants possess distinct habituation time constants

We investigated whether strains of *Drosophila* known to have learning and memory deficiencies have different habituation profiles compared to wild-type strains. Specifically: (i) the desensitization to continuous vibration, characterized by $\tau_{des}$; (ii) the re-sensitization to vibration after stimulus removal, characterized by $\tau_{res}$; and (iii) the changing re-sensitization rate after repeated pulse exposure, characterized by $\tau_{res}$ vs. $n$. Three mutant strains were tested: *rut*, lacking the Rutebaga gene; *dnc*, lacking the Dunce gene; and *cam*0, a calmodulin null mutant. We focused on the stronger, reverse crawl-aversive response, observing $F_{REV}(t)$ for each strain.

In response to continuous vibration (**Figure 6A**), all three mutant strains have habituation time constants significantly different from wild type, with *rut* the fastest adaptation ($\tau_{des} = 5.2$ s), *cam*0 the slowest (25.6 s), and *dnc* in between (14.3 s). The wild-type desensitization time (from **Figure 4B**) was 18.9 s. We note that *dnc* and *rut* mutants trend in the same direction, with both de-sensitizing more rapidly than wild type. The *dnc* mutant also has a distinct, short time scale peak in reverse crawl response, not seen in the other three strains. We also note that the baseline reverse crawl probability is very low for all three mutant strains, similar to the $\approx 0.03$ value seen for wild-type larvae. The *dnc* larvae do appear to exhibit an even lower baseline, although this is not statistically significant with the number of animals we tested here. The overall locomotion features are also very similar for all four strains, although this is not specifically tested in our work here.

As observed above (**Figure 5E**), wild-type response to repeated pulses consists of repeated shapes of $F_{REV}(t)$, but at diminished magnitude, indicating an incomplete return to the baseline level of sensitivity. We measured the recovery of vibration sensitivity for the three mutants in **Figure 6B**, and as before we calculate the ratio $F_{n,REV}/F_{n-1,REV}$ as a function of $T_{OFF}$ to extract re-sensitization times between each pair of sequential vibration pulses in the stimulus sequence. After the initial ($n = 0$) pulse, wild-type larvae recover with a time constant of $\tau_{res} = 5.3$ s, much shorter than the de-sensitization time following the initial onset of the stimulus. The three mutant strains re-sensitize with distinct time constants 3.6 s (*rut*), 6.5 s (*cam*0), and 9.8 s (*dnc*), with these times significantly different from each other, but only *dnc* significantly different from wild type. We note that *dnc* and *rut* mutants trend oppositely for re-sensitization, unlike their de-sensitization response described above. All three mutants share the feature that $\tau_{des} > \tau_{res}$, where de-habituation occurs more rapidly than habituation.

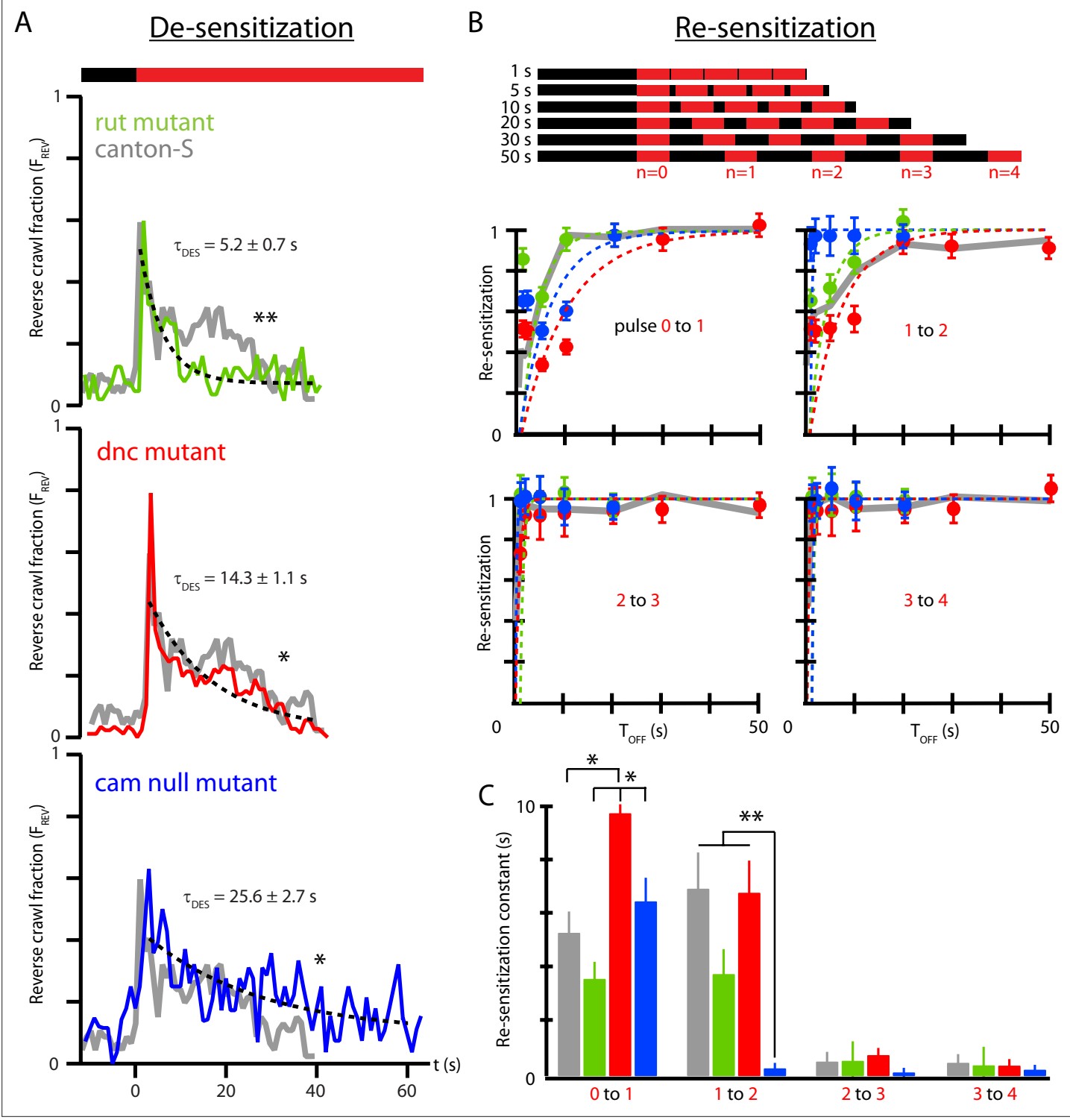

**Figure 6.** Memory-deficient mutants have distinct habituation and de-habituation time constants. (**A**) De-sensitization: reverse crawl behavior usage in response to continuous vibration stimulation. $F_{REV}$ vs. $t$ (where $t = 0$ marks the vibration onset) for three mutants: *rut* (green), *dnc* (red), and *cam0* (blue). Gray traces are the Canton-S wild-type response from *Figure 4B*. Vibrations were $f = 500$ Hz and $\Gamma = 2$. Each trace is based on five experiments, with 20 larvae in each. (**B**) Re-sensitization to vibration following repeated pulses. Top: schematic of experiments performed. Bottom: plots of $F_{n,REV}/F_{n-1,REV}$ vs. $T_{OFF}$ after the $n$th pulse for *rut* (green), *dnc* (red), and *cam0* (blue). Gray traces are the Canton-S wild-type response from *Figure 5*. Vibrations were $f = 500$ Hz and $\Gamma = 2$. Each point is based on 5 experiments, with 20 larvae in each, for a total of 75 experiments and $\approx 1500$ larvae. Error bars indicate SEM. (**C**) Desensitization and Re-sensitization time constants as functions of pulse number $n$ for the same three mutants, based on fits to the data in (**B**). Error bars indicate SEM. *p<0.05 and **p<0.01.

As with wild type (**Figure 5C**), all strains exhibit substantially faster re-sensitization after the third pulse compared to after the first and second pulses. The $\tau_{res}$ vs. $n$ relationship is shown directly for all three strains in **Figure 6C**. The *cam*0 mutants specifically show a dramatic drop in $\tau_{res}$ even after the second pulse, with nearly instantaneous recovery: the strain with the slowest habituation is the fastest to de-habituate after repeated stimulus pulses.

Put together, we find that each mutant exhibits distinct deviation from typical wild-type behavior, making it important to separate the three parameters that describe adaptation to mechanical agitation. To fully understand the molecular mechanisms behind habituation and its component time constants is beyond the scope of this article. However, these results suggest the need to describe habituation with at least these three parameters, each of which may have distinct cellular or molecular underpinnings.

### An electric circuit model is analogous to habituation

In this section, we seek to develop a visualization tool that helps conceptualize how larvae habituate and de-habituate to repeated stimulus exposures. We imagine habituation as a physical substance, electric charge on a capacitor. We use a simple electrical circuit model that is able to capture some aspects of reverse crawl response to vibration, and that suggests some possible underlying mechanisms for habituation in larvae.

Our findings so far suggest that the process underlying habituation is based on some mechanism that involves activation and recovery. The overall response of *Drosophila* larvae to vertical vibration depends on both intensity ($f, \Gamma$) and timing ($T_{OFF}, T_{ON}$) characteristics. The reverse crawl behavior is generally only seen when the vibration intensity crosses a rough threshold in $f - \Gamma$ space (**Figure 2**). We also found that the deployment of reverse crawling (measured by $F_{REV}$) decreases sharply during

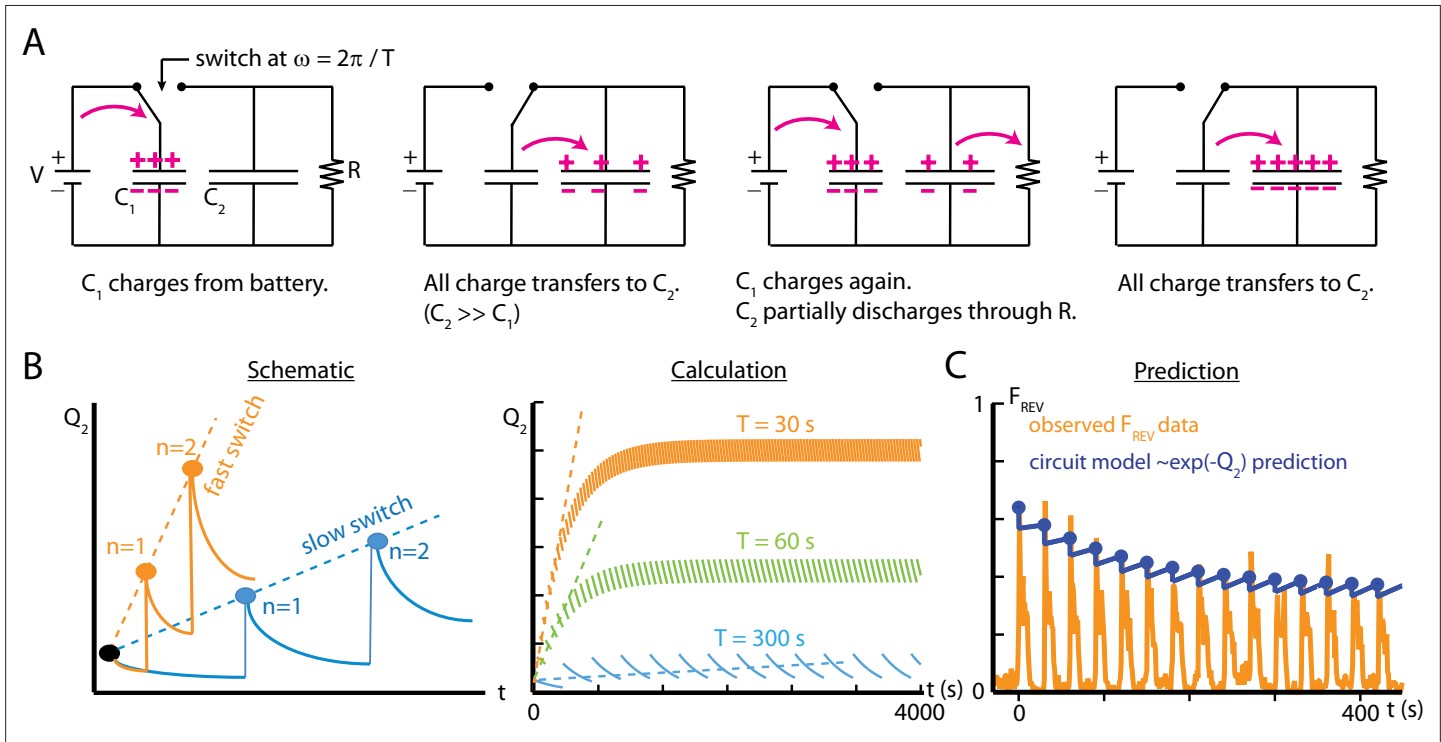

**Figure 7.** An electric circuit models possible mechanisms for larval habituation. (**A**) The capacitor switch circuit, where a small capacitor $C_1$ is continually charged by a battery $V$, and discharges to a larger capacitor $C_2$ each time the switch changes. The charge $Q_2$ is related to the probability $P$ of observing an external event ($Q_2 = -\ln P$). (**B**) Functions $Q_2(t)$ created by varying the duration of the charging phase of the circuit, $T$, while holding the circuit elements constant. Left: a visual schematic of such functions. Right: $Q_2(t)$ generated by simulating the circuit behavior. In each case, after enough switches, the charge saturates when the charging from $C_1$ to $C_2$ balances the charge dissipated through $R$ for each cycle. For values of $T$ much smaller than $RC_2$, this saturation will only occur at a large $n$. (**C**) Comparison of the circuit model to empirical data of reverse crawl probability ($f = 500$ Hz, $\Gamma = 2$, $T_{ON} = 10$ s, $T_{OFF} = 20$ s, same data shown in **Figure 4C**). The circuit values are $V = 1$, $C_1 = 1.2$, $C_2 = 160$, $R = 1$, with the switch operating with period $T = 30$. The peak values from the circuit model do not significantly differ from the empirical peaks ($p > 0.05$, Fisher's exact test).

extended or repeated vibration bouts, back toward baseline behavior (**Figure 4**). Further, we found that $F_{REV}$ returns to its original sensitivity for subsequent vibrations, dependent on the stimulus off-time $T_{OFF}$, and the vibration pulse number $n$ (**Figure 5**). We seek to establish an electric circuit model of the habituation process using a circuit with a small number of components that can reproduce the desensitization observed in behaving larvae. The electric circuit is not intended directly as a model of the neural circuit of the larva, but as a physical analogy to help visualize what appears to be occurring based on empirical behavior results presented above.

We model the situation in the larva as follows. During exposure to a stimulus, a binary process is switched on and then reset upon termination of the stimulus. The process contributes a discrete amount to a quantity $Q$, which is related to the probability $P$ that a particular behavioral output (e.g., one of the four responses shown in **Figure 1C**) will occur during the subsequent onset of the stimulus. If the frequency of these on/off switches increases, then the frequency of contributions to $Q$ also increases. If $Q$ also decays on its own over time, then the two separate mechanisms (discrete contribution to $Q$ and decay of $Q$) will together determine the overall probability of the behavioral response, similar to our observed adaptation behavior in larvae. We note that specifically $Q$ is proportional to $-\ln P$, and that $Q$ represents the level of 'habituation' in the system; when $Q = 0$, the probability of the behavior does not diminish. Using these features, we describe a capacitor switch circuit to represent a possible biological mechanism responsible for habituation in larvae.

The capacitor switch circuit is shown in **Figure 7**. Consider the circuit's behavior for the two switch positions. In the left position ('OFF'), a battery of voltage $V$ quickly charges capacitor $C_1$, which then holds charge $Q_1 = C_1 V$. In the right position ('ON') a second capacitor $C_2$ gains charge from $C_1$ each time the switch is closed. We assume $C_2 \gg C_1$, so the full amount $Q_1$ is transferred each time the switch moves to the ON position. Additionally, the charge in the second capacitor, $Q_2$, is slowly dissipated through the large resistor $R$. As a function of time, the charge $Q_2$ will depend on the frequency $\omega$ at which the switch closes (or equivalently its period $T = 2\pi/\omega$), each time delivering a discrete quantity of charge $Q_1$, and depend on the flow charge from $C_2$ through $R$. Put together, $Q_2$ will be a summation of decaying step functions

$$Q_2(t) = \sum_{n=0}^{\infty} C_1 V \theta(t - nT) e^{-\frac{t - nT}{RC_2}} \, , \tag{6}$$

where $n$ denotes the $n$th closing of the switch, and $\theta$ is a Heaviside function whose steps occur at each switch closing. We assume that $C_2$ is initially uncharged. The term $RC_2$ is a time constant describing the decay of $Q_2$.

As noted above, $Q_2$ is related to the probability $P$ of an external observed event, by $P \propto e^{-Q_2}$. Thus, the fraction of measurements where the event is observed, $F$, can be written as

$$F_{event} = F_0 \exp\left[ -\sum_{n=0}^{\infty} C_1 V \theta(t - nT) e^{-\frac{t - nT}{RC_2}} \right] , \tag{7}$$

where $F_0$ is the fraction of measurements where the event occurs when there is no charge on the capacitor $C_2$.

The capacitor switch circuit system exhibits behavior similar to what we observe empirically in larval habituation. The event fraction $F_{event}$ observed during the 'ON' switch of the circuit is analogous to the observed reverse crawl deployment fraction $F_{REV}$. The charge $Q_2$ on capacitor $C_2$ (**Equation 6**) represents a physical component of the mechanism responsible for larval habituation, such as the presence of a cytosolic concentration of a chemical or the buildup of a neurotransmitter between synapses. The repeated, discrete discharging from $C_1$ to $C_2$ is similar to the discrete contributions to desensitization caused by repeated exposure to a stimulus at some frequency; the period $T$ of such discharges determines how quickly the larvae habituate. In addition, the resistor $R$ is analogous to the recovery of the larvae, which tends to impede habituation for long time intervals, and the resistance may change over time to reflect the variation observed in $\tau_{res}$. To extend the analogy, activating the switch requires external conditions above some threshold level, and those corresponding conditions are the parameters $f$ and $\Gamma$ for mechanical agitation; below the weak vibration threshold, the reverse crawl behavior is rarely observed.

To draw a more direct comparison between the circuit model and observed crawling behavior, we characterize the response of both systems to repeated switching/vibration pulses. We compare empirical results from repeated pulses to the simple circuit model in *Figure 7C*. With suitably chosen values for the capacitances and resistance, and the time between switches matching real vibration pulse timing, the circuit model peak reverse crawl values do match the observed data, much like the de-sensitization and re-sensitization time constant modeling of *Figure 5E*. Because the circuit model delivers the full quantity of 'habituation' ($Q_1$) instantaneously at periodic intervals (as in the schematic in *Figure 7B*), the discrete addition of $Q_1$ and the decay of $Q_2$ through $R$ stand in for the whole ON and OFF parts of the cycle. This means the circuit model does not show the decline of reversal probability following each stimulus onset. We focused on building a simple model that could mirror the changes in peak reversal probabilities, rather than the dynamics between stimulus onsets.

## Discussion

This study has investigated the response to vertical vibration of the *Drosophila* larva, which deploys a range of behaviors depending on context. The severity of the response (from no response, to pausing, to turning, to reversing) reflects both the severity of the stimulus (a combination of force and frequency) and the recent history of the stimulus. Nearly all larvae stop moving upon initial exposure to high-intensity vibrations (*Figure 2B*) and use the strongest reverse crawl response in a large fraction of cases. However, we found that the reverse crawl response diminishes, and behavior returns to the non-stimulus baseline level over less than 30 s of sustained vibration. Hence, a comprehensive description of behavioral response to vibration necessarily includes time constants characteristic of adaptation: a desensitization time and a re-sensitization time (*Figures 4 and 5*). Our general characterization of vibration response, combined with our result that memory-deficient mutants exhibit anomalous de- and re-sensitization (*Figure 6*), and our electric circuit model (*Figure 7*), informs a discussion of possible mechanisms behind vibration response and habituation.

### Possible mechanisms for vibration response and habituation

In this section, we draw from our empirical results presented here, our electric circuit model, and the literature to offer suggestions about possible mechanisms underlying vibration response and its corresponding habituation characteristics. Numerous further experiments would need to be performed to establish more definitely what causes habituation to mechanical stimuli in fly larvae.

Interaction between the peripheral nervous system (PNS) and the central nervous system (CNS) should determine behavioral response to vibration. The more severe and less spontaneous reverse crawl response ($F_{REV}$), for example, could operate analogously to our circuit model (*Figure 7*), with the PNS controlling the switch and the CNS acting as the capacitor $C_2$ and mediating signals sent to the muscles. The diminished fraction of $F_{REV}$ after repeated pulses could be explained by biological processes that affect the number of signals sent to the muscles via the CNS, such as cAMP inhibition, a decrease in neuronal excitability, or both.

The fact that *dnc* mutants re-sensitize more slowly after stimulus removal may point to cAMP as important for the response process: *dunce* encodes cAMP-specific phosphodiesterase (PDE) (*Conti et al., 2003*), which breaks down cAMP and affects cAMP metabolism and synaptic plasticity (*Zhong and Wu, 1991*; *Waltereit and Weller, 2003*). The enzyme PDE thus could be important for the sensory recovery of larvae in general. Furthermore, cytosolic cAMP concentration (analogous to $Q_2$) within a subset of the CNS (analogous to $C_2$) may relate to weaker behavioral response due to habituation, similar to the circuit model like $F_{REV} \sim \exp(-[C_{cAMP}])$. Studies of memory in *Drosophila* have shown trends similar to this relationship and demonstrated effects of *dnc* on habituation to olfactory stimuli (*Engel and Wu, 2009*; *Dudai, 1988*; *van Swinderen, 2007*; *Rees and Spatz, 1989*). Dunce mutants *dnc* were used to establish the role of the cAMP cascade in neuromuscular transmission that mediates the habituated response, analogous to the discrete activation of the signaling pathway (charging capacitor $C_2$ in the circuit model) (*Zhong and Wu, 1991*). This possibility is supported by the fact that calmodulin null mutants, which lack the ability to convert ATP to cAMP in cells, exhibit an anomalous reverse crawl behavior compared to wild-type larvae (*Heiman et al., 1996*). Furthermore, *dnc*, despite being expressed throughout neuropil, is concentrated in mushroom body (MB) neurons (*Nighorn et al., 1991*; *Han et al., 1996*) and studies investigating olfactory habituation in larvae point to the

alteration in the excitability of postsynaptic MB neurons as crucial to the process. MB neurons could play a role in habituation behavior for mechanosensation (*Davis, 1993*; *Engel and Wu, 2009*; *Hollis and Guillette, 2011*; *Neckameyer, 1998*), which would indicate a significant crossover between the neural mechanisms responsible for mechanosensitive and olfactory habituation.

Past studies investigating a similar mechanical response in *C. elegans* have established that the mechanism responsible for habituated behavior depends on interactions between PNS neurons and proprioceptor neurons in the CNS (*Stopfer and Carew, 1996*; *Rosen et al., 1979*; *Rose and Rankin, 2001*). These neurons correspond to dendritic and chordotonal neurons respectively in *Drosophila* larvae (*Tuthill and Wilson, 2016*). Given that cAMP-signaling cascades and neural excitability have been established as important processes related to the short-term plasticity of chordotonal neurons in general (*Waltereit and Weller, 2003*; *Zhong and Wu, 1991*), it is possible that the mechanosensitive habituated response mechanism in larvae is dependent on processes at the postsynapse of these neurons, in a manner similar to habituation in *C. elegans* (*Bozorgmehr et al., 2013*). Thus, a possible explanation for mechanosensitive habituation in larvae is the activation of postsynaptic ion channels during stimulation, specifically, voltage-dependent potassium ion channels modulated by neurotransmitter signaling at the postsynapse of motor neurons. These ion channels could significantly decrease the neuronal excitability of the motor neuron to which they are attached. If a subset of these motor neurons in *Drosophila* are involved in the circuit for reverse crawling, then activation of the ion channels would decrease the likelihood of a 'reverse crawl signal' sent by a neuron, and thus decrease the probability the behavior is performed. Such a mechanism has been identified in the mechanosensory circuit of *C. elegans* (*Bozorgmehr et al., 2013*) and is a promising candidate in *Drosophila* since it could more effectively account for the dependence of re-sensitization on $T_{OFF}$. In addition, the mechanism is most analogous to the capacitor switch circuit model, whereby calcium ions act as the charge $Q_2$ and the inter-neural channel acts as $C_2$. As neurons reset following action-potential activation, the calcium concentration in the region is slowly reduced, whereas the amount of calcium added is dependent on the discrete activation of presynaptic dendritic neurons. GABA, which has been identified as crucial for larval olfactory habituation (*Larkin et al., 2010*) and shown to bind to input sites on other invertebrate chordotonal neurons (*Panek et al., 2002*; *Cattaert et al., 1992*; *Burrows and Laurent, 1993*), could potentially regulate the activation threshold of the described ion channels. Other types of neurotransmitters, such as glutamate or dopamine, may also play a role in larval mechanosensitive habituation in chordotonal neurons.

## Conclusions

In our investigation of the *Drosophila* larva's response to vertical vibration, we have particularly focused on the deployment of discrete physical motor actions, and how the animal's use of each behavior changes over time due to habituation. We found that adaptation is a very strong effect, shown by the LTI model's failure to capture the empirical response. Because these experiments captured both population-level and single-larva movement, we were able to confirm that transitions between behavioral states closely approximate a one-way habituation model, where weaker avoidance behavior replaces stronger behaviors, and individual animals will very rarely reverse crawl after switching to a milder response. Three adaptation parameters were necessary to account for the response to a sequence of vibration pulses: a desensitization time scale ($\tau_{des}$) for a continued stimulus, a re-sensitization time scale ($\tau_{res}$) for robustness to return in the absence of the stimulus, and the shortening of $\tau_{res}$ after repeated pulses. We gained insight into potential mechanisms behind this highly adaptive response, first through behavior experiments with larval mutants, which exhibited distinct variations in the three adaptation parameters compared to wild type, then through comparison with our charge transfer electric circuit model, which appears to map to distinct parameters of the observed behavior in a manner indicative of information retention producing an altered behavioral output in larvae.

We note several limitations of the present work. First, because we deployed only the Canton-S wild-type larva strain for nearly every behavior experiment, and not other wild-type strains or rescue experiments with shared genetic backgrounds, strict conclusions about the behavior of genetic mutants (*rut*, *dnc*, *cam*) should be avoided. The mutant results of *Figure 6*, while interesting and suggestive, are there to highlight the importance of determining multiple time constants and other dependencies when characterizing habituation in this system. Further control experiments with rescue strains would be necessary as a starting point to better understand the specific roles and functions of these genes.

Several directions for further study are apparent. Because the animal's response to vertical vibration depends on both the vibration's severity (force and frequency) and its recent history (number of pulses and ON/OFF times in our framework), the parameter space for a complete mapping of stimulus input to behavioral output is very large. A combination of improved hardware to explore a larger range of input conditions and novel stimulus delivery (such as noise stimulus with reverse correlation analysis) could cover a broader range of responses and generate more directly testable mathematical functions that predict probabilities of each behavior. How vibration combines with other sensory inputs to produce a multisensory integration output is also an interesting question, especially because vibration response is highly nonlinear and dominated by habituation, whereas many other stimuli yield more straightforward responses. Finally, because the fly larva is such an optically and genetically addressable system, interrogating the neural circuits involved in adaptation should prove fruitful. For temperature, odors, and other stimuli, optical calcium or voltage imaging of the sensory neurons and central brain can be performed during stimulus delivery, and a miniature version of the vibration system used here could allow the same for vertical vibration. Because habituation forms so quickly in the larva, the system should be ideal for monitoring desensitization and re-sensitization in the brain in real time.

Understanding the biological process responsible for mechanosensitive habituation in larvae is an area for potential continued research. This study has investigated a few important aspects of the habituated behavior in larvae and has shown that these observations are indicative of a process that employs neural mechanisms on very short time scales to induce plasticity. The neurophysiological and biological processes that take place within *Drosophila* larvae to cause habituation are, in general, suited to the organism's most general purpose of survival and may serve a wider role in the survival of more complex organisms that must navigate random and complex natural environments. Mechanical agitation is a useful stimulus for attempting to decipher the habituation phenotype and its underlying mechanisms.

## Materials and methods

### Vertical vibration and image acquisition

The top piece of an EMT (ET-132-203, LabWorks Inc) is displaced upward and downward. An aluminum plate ($230 \times 230 \times 1.8$ mm) with a hole drilled in the center was placed atop a steel damping plate ($150 \times 150 \times 5$ mm), also with a hole drilled in the center. These two plates were then screwed into the top of the EMT. The steel plate reduced the strength of vibrational nodes in the system. The EMT was placed atop a 3-mm-thick rubber sheet to prevent the migration of the device during testing.

The EMT was driven by a sine wave controller (SG-135, LabWorks Inc) and an amplifier (PA-151, LabWorks Inc) that provided AC current up to 2.5 A at the frequency specified by the controller. A small accelerometer with a flat end was used to measure the peak acceleration of the agar gel placed on the aluminum plate at various locations (20–30 points), both for calibration and to determine spatial variation in $\Gamma$. The typical variation was $< 0.1$, with maximum variation $\delta\Gamma \approx 0.3\Gamma$ only observed at low frequencies.

The connection between the power amplifier and the EMT was interrupted by a solid-state relay (4D1225, Crydom) to allow for computer control of the ON and OFF states of vibration pulses, via a USB DAQ device (U3-LV, LabJack) Using custom software written in LabView, the vibration signals were sent to the EMT according to the desired $T_{ON}$ and $T_{OFF}$ timing.

The EMT was placed within a sealed box along with four printed circuit boards (PCBs) with red LEDs and a camera directly over the crawling surface. Each PCB had 48 lights, with 12 sets of four lights and a current regulator. The LED boards were held in place by custom PLA stands made by a 3D printer (Ultimaker 2) and powered by a 12 V DC power supply (SE-350-12, Meanwell). The LEDs were held slightly above the gel surface, facing inward, to provide dark field illumination of the crawling animals.

A 5 MP CCD camera (acA2500-14, Basler) was attached to the top beam of the box. Image acquisition software (same as used in *Gershow et al., 2012*) was modified to synchronize with the vibration control software, so vibration pulse sequences matched the timing of the behavior recordings. Typically we recorded 90 s of behavior prior to the first vibration period. The images were recorded at 15 frames per second.

## Data analysis

We used a modified version of the MAGAT Analyzer, which determines the position and contour of each larva throughout a recording, segments trajectories into straight-crawling 'runs' and reorienting 'turns,' and determines numerous parameters like velocity, body bend angle, and so on *Gershow et al., 2012*. Custom MATLAB scripts flagged the four primary response behaviors of interest here (continuation, pause, turn, reversal). We computed the dot product of the head orientation vector and the velocity vector, with negative values indicating reverse crawling.

Curve fits characterizing habituation were performed by fitting $F_{REV}(t)$ data to the function $y_0 + A \exp(-t/\tau)$ for both desensitization and re-sensitization, with $y_0$ fixed to be the baseline $F_{REV}$ value and the other parameters free. Uncertainty in the fits, and comparison between different fits, was determined using the following steps: (1) a simulated value of $F_{REV}$ at each time point (1 s spacing) was pulled from a Gaussian distribution centered at the mean value with the SEM as the width, and the exponential fit was performed on this generated set of points; (2) this step was repeated 1000 times, and the standard deviation of the set became the uncertainty of the original curve fit; and (3) significance tests between different exponential fits (e.g., the wild type vs. mutant strains) were performed as standard Student's *t*-tests, using the set of 1000 fit values, but with the z-scores using standard deviation instead of SEM, obtained by multiplying the calculated z-score by $\sqrt{1000}$ (otherwise the number of simulated fits would affect statistical significance). The p-values in *Figure 6* are denoted with * symbols explained in the caption. Actual values comparing desensitization time constants in *Figure 6A* are p<0.001 (*rut*/CS), p=0.011 (*dnc*/CS), and p=0.026 (*cam0*/CS). p<0.0001 for all pair-wise comparisons between the three mutant strains. Actual values comparing re-sensitization time constants in *Figure 6C* are p=(*cam0*/CS), 0.007 (*cam0*/*rut*), 0.003 (*cam0*/*dnc*), 0.07 (*rut*/CS), 0.0001 (*dnc*/CS), and < 0.0001 (*rut*/*dnc*) for the first re-sensitization; then p=(*cam0*/CS), 0.0002 (*cam0*/*rut*), < 0.0001 (*cam0*/*dnc*), 0.23 (*rut*/CS), 0.98 (*dnc*/CS), and 0.03 (*rut*/*dnc*) for the second re-sensitization.

No explicit power analysis was used to compute sample size in the initial design of our study, but we recorded repeated experiments until the fractional SEM was small. The most common number of 100 animals per experimental condition was more than sufficient to distinguish most behavioral differences, consistent with prior work in fly larva behavior. Most commonly 20 larvae were placed together on the gel in the vibration arena for each experiment, which balances high throughput with larva–larva interactions becoming too frequent, and is commonly used in arenas of this size. Occasional human error in counting, or immobile animals, or animals with repeated collisions were encountered, so the exact number of tracks analyzed was not always known, but we estimate that the number of animals in each experiment was always between 18 and 22. The behavior of any moving larva was included in every analysis of every experiment.

## *Drosophila* handling

Canton-S wild-type adult flies were kept in cages (Genesee Scientific) with 6 cm Petri dishes with grape juice and yeast food, with new plates exchanged every 24 hr. Animals were collected from the plates, selecting second-instar larvae by age (24–72 hr AEL) and spiracle development of each individual, but otherwise randomly chosen. The typical larva size at this instar is 1–2 mm in length. For each experiment, between 20 and 25 larvae were rinsed in distilled water, allowed to crawl on agar gel (3% wt./vol) for 5 min, then placed on a separate dark agar gel atop the aluminum plate of the EMT. The mutant strains were treated the same way. Larvae were discarded after each single experiment, and not used again.

All animals for the experiment were placed on the agar surface together, near the center, with approximately 1 cm separating each animal. Given the small fraction of the available space taken up by the animals, collisions were infrequent. Importantly, when a collision does occur, the event is not flagged as a turn for the purposes of avoidance behavior computation; so if the collision rate decreases over time as animals spread out, the extracted information is unaffected.

## Acknowledgements

The authors thank James Baker and Kevin Collins for comments on the manuscript.

## Additional information

### Funding

| Funder | Grant reference number | Author |
|---|---|---|
| National Science Foundation | 2144385 | Mason Klein |

The funders had no role in study design, data collection and interpretation, or the decision to submit the work for publication.

### Author contributions

Alexander Berne, Conceptualization, Data curation, Software, Formal analysis, Investigation, Visualization, Methodology, Writing - original draft, Writing - review and editing; Tom Zhang, Conceptualization, Resources, Methodology; Joseph Shomar, Conceptualization, Methodology; Anggie J Ferrer, Conceptualization, Resources, Supervision; Aaron Valdes, Investigation; Tomoko Ohyama, Conceptualization, Writing - review and editing; Mason Klein, Conceptualization, Resources, Software, Formal analysis, Supervision, Visualization, Project administration, Writing - review and editing

### Author ORCIDs

Alexander Berne  http://orcid.org/0000-0001-7857-8513
Joseph Shomar  https://orcid.org/0009-0000-9534-6263
Tomoko Ohyama  http://orcid.org/0000-0003-1697-1138
Mason Klein  http://orcid.org/0000-0001-8211-077X

### Decision letter and Author response

Decision letter https://doi.org/10.7554/eLife.69205.sa1
Author response https://doi.org/10.7554/eLife.69205.sa2

## Additional files

### Supplementary files

• Transparent reporting form

• Source code 1. Matlab script, which should be run in the MAGAT Analyzer environment after loading movie files, that determines the behavioral state of crawling larvae, in particular identifying reverse-crawl behavior.

• Source code 2. Functions in Igor Pro that perform fits to larva crawling data, used to extract desensitization and re-sensitization time constants.

### Data availability

Analysis code files have been provided for all figures. Raw behavior data has been made available on Harvard Dataverse (https://doi.org/10.7910/DVN/9JWPN2).

The following dataset was generated:

| Author(s) | Year | Dataset title | Dataset URL | Database and Identifier |
|---|---|---|---|---|
| Klein M | 2023 | Crawling Trajectories of *Drosophila* Larvae Responding to Vibration | https://doi.org/10.7910/DVN/9JWPN2 | Harvard Dataverse, 10.7910/DVN/9JWPN2 |

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
