## [Editor Report]

This is a strong article due to its sophisticated behavioral analysis and modeling of behavioral output, and the system and results provide a framework for future genetic analysis examining the biological basis of sensory behaviors.

---

## [Decision Letter]

**Decision letter after peer review:**

Thank you for submitting your article "Mechanical vibration patterns elicit behavioral transitions and habituation in crawling *Drosophila* larvae" for consideration by eLife. Your article has been reviewed by 3 peer reviewers, and the evaluation has been overseen by a Reviewing Editor and K VijayRaghavan as the Senior Editor. The following individuals involved in the review of your submission have agreed to reveal their identity: Jan Clemens (Reviewer #2).

Essential revisions:

*Reviewer #2 (Recommendations for the authors):*

1) The authors use diverse modeling approaches to provide insight into the behavior - Markov models (Fig. 3), impulse responses (Fig. 4), a phenomenological model based on time constants (Fig. 5, 6), an electrical circuit model (Fig. 7). I really like this approach and I understand that each approach serves a specific purpose. However, this diversity of models also makes it hard to link different results - see my comment #2 on the Markov models. Moreover, each model comes with its own metaphor/analogy and having to mentally switch between the different approaches and analogies can be challenging for readers that are not familiar with the computational modelling. For instance, the authors analyze mutants for dnc, rut, and cam and I did then expect a model that is based on abstract molecular quantities. However, the model that is used to explain the findings is first introduced in terms of abstract quantities Q (lines 306-315), which are then revealed to be capacitors C (lines 317-331) which in turn are interpreted as concentrations of chemicals (lines 332-335). I understand the reasoning behind these decisions (electrical circuit models are standard in the modeling community) and this may be impossible to change. But a more direct approach may make the paper more accessible to the wide audience it deserves.

2) The plots that show re-sensitization (for instance, Fig. 5B, 6B) are hard to read due to the fast dynamics. Using a log scale might make the complex dynamics of re-sensitization more obvious. For instance, the re-sensitization seems to start high and then dip for the first steps before recovering (Fig. 6B).

---

## [Author Response]

Essential revisions:Reviewer #2 (Recommendations for the authors):1) The authors use diverse modeling approaches to provide insight into the behavior - Markov models (Fig. 3), impulse responses (Fig. 4), a phenomenological model based on time constants (Fig. 5, 6), an electrical circuit model (Fig. 7). I really like this approach and I understand that each approach serves a specific purpose. However, this diversity of models also makes it hard to link different results - see my comment #2 on the Markov models. Moreover, each model comes with its own metaphor/analogy and having to mentally switch between the different approaches and analogies can be challenging for readers that are not familiar with the computational modelling. For instance, the authors analyze mutants for dnc, rut, and cam and I did then expect a model that is based on abstract molecular quantities. However, the model that is used to explain the findings is first introduced in terms of abstract quantities Q (lines 306-315), which are then revealed to be capacitors C (lines 317-331) which in turn are interpreted as concentrations of chemicals (lines 332-335). I understand the reasoning behind these decisions (electrical circuit models are standard in the modeling community) and this may be impossible to change. But a more direct approach may make the paper more accessible to the wide audience it deserves.

This was a rather difficult paper to write, for the reasons the referee mentions. We are happy that the approaches are appreciated, and agree that it can be hard to link everything narratively in a way that doesn’t mislead or confuse the reader. We have overhauled our writing about the circuit model, and have also added to the introductions to some sections, stating ahead of time what the goal is and what we will find. The beginning of the circuit sub-section in particular is more clear about what will be accomplished. We hope this makes the results flow much better.

2) The plots that show re-sensitization (for instance, Fig. 5B, 6B) are hard to read due to the fast dynamics. Using a log scale might make the complex dynamics of re-sensitization more obvious. For instance, the re-sensitization seems to start high and then dip for the first steps before recovering (Fig. 6B).

We thought about also showing them that way for our initial submission, but opted to go with the linear scale only at the time. An example log version (of 1 – re-sensitization) is included in Author response image 1. It was feeling like too many graphs for Figs. 5 and 6, but we would certainly be OK with adding these if the editors/reviewers feel it would be informative.